# Marine Cyanobacterial Peptides in Neuroblastoma: Search for Better Therapeutic Options

**DOI:** 10.3390/cancers15092515

**Published:** 2023-04-27

**Authors:** Salman Ahmed, Waqas Alam, Michael Aschner, Rosanna Filosa, Wai San Cheang, Philippe Jeandet, Luciano Saso, Haroon Khan

**Affiliations:** 1Department of Pharmacognosy, Faculty of Pharmacy and Pharmaceutical Sciences, University of Karachi, Karachi 75270, Pakistan; 2Department of Pharmacy, Abdul Wali Khan University Mardan, Mardan 23200, Pakistan; 3Department of Molecular Pharmacology, Albert Einstein College of Medicine Forchheimer, 209 1300 Morris Park Avenue, Bronx, NY 10461, USA; 4Department of Science and Technology, University of Sannio, 82100 Benevento, Italy; 5State Key Laboratory of Quality Research in Chinese Medicine, Institute of Chinese Medical Sciences, University of Macau, Macao SAR 999078, China; 6Faculty of Sciences, RIBP-USC INRAe 1488, University of Reims, 51100 Reims, France; 7Department of Physiology and Pharmacology “Vittorio Erspamer”, Sapienza University, 00185 Rome, Italy

**Keywords:** marine cyanopeptides, apoptosis, autophagy, cell cycle arrest, antimetastatic

## Abstract

**Simple Summary:**

This review describes the anticancer activities of several marine peptides isolated from cyanobacteria and their specific effects against neuroblastoma. After a short presentation of the major cyanobacterial peptides of marine origin, mainly belonging to the cyclic depsipeptide family, this work has focused on the different mechanisms of action of these peptides (effects on apoptosis, cell cycle arrest, autophagy, sodium channel blocking as well as antimetastatic activities). A thorough description of the biological effects of the marine cyanobacterial marine peptides is developed including data on their half-maximal effective concentration (EC_50_), growth inhibition 50% (GI_50_) and lethal concentration 50 (LC_50_) on various neuroblastoma cell lines. This review ends with a description of the clinical trials which are underway to evaluate the anticancer effects of peptides arising from marine cyanobacteria and microalgae. Conclusions envisage the potential role of such peptides for the development of anti-neuroblastoma medicines and a platform for uncovering new therapeutic cellular targets.

**Abstract:**

Neuroblastoma is the most prevalent extracranial solid tumor in pediatric patients, originating from sympathetic nervous system cells. Metastasis can be observed in approximately 70% of individuals after diagnosis, and the prognosis is poor. The current care methods used, which include surgical removal as well as radio and chemotherapy, are largely unsuccessful, with high mortality and relapse rates. Therefore, attempts have been made to incorporate natural compounds as new alternative treatments. Marine cyanobacteria are a key source of physiologically active metabolites, which have recently received attention owing to their anticancer potential. This review addresses cyanobacterial peptides’ anticancer efficacy against neuroblastoma. Numerous prospective studies have been carried out with marine peptides for pharmaceutical development including in research for anticancer potential. Marine peptides possess several advantages over proteins or antibodies, including small size, simple manufacturing, cell membrane crossing capabilities, minimal drug–drug interactions, minimal changes in blood–brain barrier (BBB) integrity, selective targeting, chemical and biological diversities, and effects on liver and kidney functions. We discussed the significance of cyanobacterial peptides in generating cytotoxic effects and their potential to prevent cancer cell proliferation via apoptosis, the activation of caspases, cell cycle arrest, sodium channel blocking, autophagy, and anti-metastasis behavior.

## 1. Introduction

The most common type of extracranial solid tumor in pediatric patients is neuroblastoma (NB), which is derived from the cells of the sympathetic nervous system. It is most common in the abdomen area, particularly around the adrenal glands, but can also appear in nerve tissues of the neck, chest, abdomen or pelvis. Neuroblastoma is diagnosed in children under two with a poor prognosis. In almost 70% of patients, metastasis can be seen after diagnosis. Conventional treatments for NB include chemotherapy (e.g., Cyclophosphamide or Dinutuximab), radiotherapy, immunotherapy (e.g., CAR T cell- chimeric antigen receptor T cell), and surgical tumor resection. Dinutuximab is a chimeric antibody directed against GD2 present on neuroblastoma cells and used as an immunotherapeutic agent in selected neuroblastoma patients. Despite the wide range of cancer therapies, current therapies do not provide optimal results since cancer recurrence and metastasis are prevalent. NB thus remains a significant problem with a high mortality rate [1,2].

Cancer chemotherapy is undergoing a dramatic transformation as a growing number of targeted drugs increase the therapeutic efficacy, minimize destructive impacts, and improve health outcomes. Despite recent advances in chemotherapy, the prognosis of advanced NB remains poor, and its treatment is associated with adverse side effects (toxicity and myelosuppression). Surgical procedures are invasive and may result in inadequate tumor excision, requiring additional chemo- and radiation therapies and stem cell transplantation. Therefore, it is crucial to discover more selective chemicals for cancer treatment with fewer adverse effects, more robust therapeutic efficacy, and a reduced resistance index. Increased progress is underway to obtain effective naturally sourced chemicals. Accordingly, new anticancer medicines with minimal adverse effects are needed for the effective treatment of NB [3,4,5].

Natural compounds, mainly from marine organisms (microbes and plants), have been widely studied as complementary and supportive therapies for cancers aimed at affording a preventative role in cancer care, reducing the adverse effects of oncologic treatments, and overcoming cancer drug resistance [6,7,8].

Marine medicines constitute an essential source of anticancer treatments. Despite the vast potential of new marine drugs, only a few pharmaceuticals have been used for cancer treatment to date. Following the initial acceptance of cytarabine in 1969, the FDA approved several marine-derived compounds as anticancer drugs. The discovery of ulithiacyclamide, the first maritime antitumor peptide, was followed by other marine anticancer peptides, such as didemnin B, dolastatin 10, kahalalide F, hemiasterlin, cemadotin, soblidotin, aplidine, and others, with subsequent clinical trials [9,10,11,12]. Marine cyanobacteria have aroused considerable interest in marine ecology due to their abundance and ability to provide novel chemotypes with substantial biological activity.

*Nostoc* sp. was initially used to cure gout, fistula, and other malignancies in approximately 1500 BC. However, in the 1990s, a more concerted effort was initiated in this area [13,14,15]. Numerous prospective studies have been conducted with marine peptides for pharmaceutical development including anticancer potential. Marine peptides have several advantages over proteins or antibodies, including their small size, simple manufacturing, cell-membrane-crossing capabilities, minimal drug–drug interactions, minimal changes in blood–brain barrier (BBB) integrity, selective targeting, chemical and biological diversities, and effects on liver and kidney functions [13,14,15]. Anticancer peptides have short half-lives, limited bioavailability, poor pharmacokinetic parameters, first-pass metabolism, and sensitivity to proteases. Peptides are classified as apoptosis inducers, cell proliferation and angiogenesis blockers, antioxidants, microtubule-destabilizing agents, to name a few [16,17,18,19,20]. Peptides constitute the majority of secondary metabolites of cyanobacteria. Peptides of several structural types have been identified including linear and cyclic depsipeptides as well as lipopeptides with various multidimensional anticancer mechanisms [21,22,23]. Cyclopeptides/cyclic peptides have piqued the interest of marine natural product researchers as a potential way for drug evolution because of their high binding affinity, target selectivity, low toxicity, effective penetration of tumors, enhanced resistance to exo- and endopeptidase degradation and increased bioavailability in vivo. These properties outperform linear peptides and other compounds for therapeutic applications [24,25,26,27,28]. Cyclo depsipeptides possess a more complex structure where consecutive ester bonds replace additional amide bonds [29]. Aurilides B-C [30], desmethoxymajusculamide C [31], guineamides [32,33], tiahuramides [34], palmyramide A [35], hoiamide A [36], lyngbyabellins E–I [37] from *Lyngbya majuscula*; coibamide A [38], grassypeptolides and ibu-epidemethoxylyngbyastatin 3 [39] from *Leptolyngbya* sp.; symplocamide A [40], largazole [41] from *Symploca* sp.; bouillonamide, ulongamide A, apratoxin A [42] from *Lyngbya bouillonii* are some examples of cyclic depsipeptides displaying anti-NB effects. Lipopeptides are amphiphilic compounds with hydrophobic and hydrophilic fatty acid chains and cyclic peptides, respectively [43,44]. Hermitamides [45], jamaicamides [46], malyngamides [47], somocystinamide A [48] from *Lyngbya majuscula*; dragonamide [49] and microcolins [50] from *Lyngbya *polychroa** are anti-NB peptides. A straight chain of amino acids linked by amide bonds forms acyclic or linear peptides [51]. Gallinamide A from *Schizothrix* sp. and desacetylmicrocolin B from *Lyngbya* cf. *polychroa* are reported as anti-NB cyanobacterial peptides. These peptides are active against mouse (Neuro-2a, N-18) and Human (IMR-32, NB7 and SH-SY5Y) cell lines. Anti-neuroblastoma peptides are summarized in Table 1 [52,53].

This review aims to discuss distinct natural products focusing on anti-NB cyanobacterial peptides.

## 2. Marine Cyanobacterial Peptides

Cyanobacteria, which are among the oldest aquatic and photosynthetic oxygenic prokaryotes, are found worldwide. The presence of numerous bioactive secondary metabolites in cyanobacteria from various habitats, especially marine cyanobacteria, has recently been discovered. Bioactive compounds from aquatic cyanobacteria help them better adapt to a variety of complex, hypersaline, high-pressure, barren marine habitats by acting as chemical defenses. These cyanobacterial secondary metabolites exhibit a wide range of biological activities, including anti-tumor, antibacterial, enzyme inhibition, parasite resistance, anti-inflammatory, and other biological activities, in addition to having a significant impact on the growth and reproduction of cyanobacteria [63]. As a result, they received interest from scholars in various experimental fields, including medicinal chemistry, pharmacology, and marine chemical ecology [64].

Over 400 new natural compounds from marine cyanobacteria have been identified over the past decade thanks to the International Cooperative Biodiversity Group (ICGB) program [65]. Peptides and compounds containing peptides are the main secondary metabolites among these substances [66]. A total of 126 novel peptide compounds, mostly from the genera *Lyngbya, Oscillatoria,* and *Symploca,* were extracted from marine cyanobacteria by the end of 2016. Nevertheless, two new genera, *Moorea* and *Okeania*, previously recognized as the polyphyletic cyanobacterial genus *Lyngbya*, were identified by genome sequence analysis [67,68]. A second new genus of *Caldora* was known as *Symploca*.

The majority of the cyclic peptides found in marine cyanobacteria are cyclic depsipeptides, which include 76 different molecules [69]. Two linear depsipeptides known as grassystatins A and B, have been isolated from the key Largo collected marine cyanobacterium *Okeania lorea* (formerly *Lyngbya* cf. *confervoides*). Veraguamides K and L, two linear bromine-containing depsipeptides isolated from the marine cyanobacterium cf. *Oscillatoria margaritifera* found in the Coiba Island National Park in Panama, are thought to have the structural characteristics of marine natural products [70]. The antimalarial bioassay-guided isolation of the marine cyanobacterium *Moorea producens* (formerly *Lyngbya majuscula*) yielded four lipopeptides: dragonamides A and B, carmabin A, and dragomabin [49]. Through the cytotoxicity-directed isolation of a marine cyanobacterium, the *Symploca* cf. *hydnoides* sample from Cetti Bay (Guam), seven novel cyclic hexadepsipeptides, known as veraguamides A–G, were discovered [71,72]. HT29 colorectal adenocarcinoma and HeLa cell lines exhibited moderate-to-mild cytotoxicity in response to these compounds [73]. *Lyngbya majuscula* has been proven to be a highly prolific species of cyanobacterium since a significant number of natural products with a wide range of structural characteristics have been isolated from it. The antimycobacterial cyclodepsipeptides known as pitipeptolides C–F were discovered from the marine cyanobacterium *Lyngbya majuscule* in the Piti Bomb Holes (Guam) [74]. Hoiamide A is an unusual cyclic depsipeptide that was isolated from the marine cyanobacteria *Lyngbya majuscula* and *Phormidium gracile* in Papua New Guinea. It is composed of an isoleucine moiety that has been modified by acetate and *S*-adenosyl methionine, a tri-heterocyclic fragment which contains two α-methylated thiazolines and one thiazole ring. This peptide can trigger the sodium inflow (EC_50_ = 2.31 M) and is a powerful voltage-gated sodium channel inhibitor (IC_50_ = 92.8 nM) in murine neocortical neurons [75]. The rapid growth, genetic tractability of cyanobacteria as well as the ease of culturing make them excellent candidates for sustainable sources for the manufacture of bioactive peptides. Although cyanobacteria share many of the same characteristics as microbes, they have received less attention.

## 3. Mechanistic Insights

### 3.1. Apoptosis

Apoptosis is an essential mechanism of cell death induced by cancer therapy. Therefore, identifying or developing anticancer agents capable of targeting apoptosis regulatory genes is a prerequisite for the advancement of unique anticancer therapies. As with most anticancer agents, there are a large number of marine-derived anticancer peptides with apoptotic activity in cancer cells [76,77]. The discharge of cytochrome-c (cyt c) activates caspases and triggers apoptosis [78,79]. Cyclolaxaphycins B and B3 increase caspase 3 in SH-SY5Y lines with IC_50_ of 1.8 and 0.8 µM, respectively (Figure 1) [61]. Coibamide A induces apoptosis in Neuro-2a cells (IC_50_ < 23 nM) through caspase-3,7 activation, cyt-c release and PARP cleavage. In U87-MG and SF-295 glioma cells, coibamide A triggered caspase-3/7 activation over a time-period associated with a loss of viability, although the activation profile for each cell line was different. Despite the fact that the MTT cell viability experiments showed that U87-MG cells were more sensitive than SF-295 cells to coibamide A-induced cell death, relatively large doses of coibamide A were required to cause the late activation of caspase-3/7 in these cells. Over a 96 h exposure period, researchers collected attached and detached coibamide A-treated cells and examined cell lysates for the expression of PARP1, a critical downstream target of caspase-dependent apoptosis, as well as a number of alternative cell death pathways [38,57,58]. Caspase-8 is one of the critical members of apoptosis initiation. Its silencing increases and facilitates NB tumorigenesis [80,81]. Somocystinamide A stimulates caspase-8 activation in Neuro-2a and NB7 cells with IC_50_ of 1.4 μg mL^−1^ and 810 nM [48,62].

Optic atrophy 1 (OPA1) is a crucial molecule in cancer cell biology and therapeutic resistance. OPA1 determines the mitochondrial resistance to cytochrome c release and delays apoptosis. The induction of OPA1 is required to alter gene expression during angiogenesis. It has also been identified as an essential regulator of lymphangiogenesis. Due to its dual role in angiogenesis and lymphangiogenesis, targeting OPA1 not only inhibits tumor development but also metastatic spread [82]. Prohibitin (PHB) is an oncogene that promotes the proliferation and differentiation of NB cells. PHB deficiency increased NB cell differentiation and death and delayed cell cycle progression [83]. Aurilide binds and inhibits prohibitin 1 (PHB1), promoting the OPA1 proteolytic processing, resulting in mitochondrial apoptosis [54,84].

Musashi-2 (MSI2) is expressed in NB, and its decreased expression is related to increased apoptosis and decreased proliferation [85]. The inhibition of the histone deacetylase (HDACi) causes apoptosis induction, PARP cleavage, and G1 or G2/M cell cycle arrest in NB cells [86]. Largazole inhibits IMR-32 cell proliferation with a GI_50_ (growth inhibitory power of the test agent) of 16 nM and SH-SY5Y with an IC_50_ of 102 nM by decreasing the MSI2 levels, suppressing the mTOR pathway, and HDACi [41,59,60]. Figure 2 summarizes the structures of therapeutically active marine cyanobacterial peptides against neuroblastoma cell lines.

### 3.2. Cell Cycle Arrest

Cell cycle interruption prevents cancer cells from developing into tumor cells and spreading to other parts of the body [87]. Grassypeptolides D and E, ibu-epidemethoxylyngbyastatin 3, and dolastatin 12 from *Leptolyngbya* sp. induce G2/M phase arrest in Neuro-2a cancer cells [39]. Similarly, coibamide A from *Leptolyngbya* sp. induce G1 to S phase arrest [58].

### 3.3. Sodium Channel Blocking Activity

The voltage-gated sodium channel (VGSC) is widely expressed in breast, bowel, prostate cancers, melanoma and NB. Several VGSCs-blocking drugs have been shown in preclinical models to limit cancer cell proliferation, invasion, tumor development, and metastasis, indicating that VGSCs may serve as putative molecular targets for cancer therapy [88,89]. Malyngamides C, J, and K were shown to block VGSCs in Neuro-2a cells displaying IC_50_ 0.49–4 μg mL^−1^ [47]. Similar effects were seen in the same cells when treated with palmyramide A with an IC_50_ of 17.2 μM [35].

### 3.4. Antimetastatic Activity

Microfilaments play an essential role in cell migration. The inhibition of actin polymerization disrupts microfilaments, reduces the cell motility, and slows the metastatic spread of neoplastic cells by G2/M phase arrest [90,91]. Microtubules and microtubule-associated proteins, which play a vital role in cell division, are essential constituents of the mitotic spindle. Microtubule dynamics is necessary for chromosomal movement throughout anaphase. A shift in the tubulin-microtubule balance alters the mitotic spindle, disrupting metaphase-anaphase progression of the cell cycle, resulting in cell death [92,93]. Microtubule-stabilizing compounds stimulate microtubule polymerization and, by binding to microtubules, target the cytoskeleton and spindle machinery of tumor cells, thus limiting mitosis [92,94]. Aurilide B-C, a cyclodepsipeptide isolated from *Lyngbya majuscula*, has been shown to destabilize microtubules in Neuro-2a cells with an IC_50_ of 0.01 and 0.05, μM, respectively [30]. Microtubule-destabilizing agents can also cause apoptosis through Bcl2 and myeloid cell leukemia-1 (Mcl-1) inhibition. Mcl-1 promotes cell survival by disrupting cytochrome-c release [95,96]. Desmethoxymajusculamide C shows efficacy against Neuro-2a cancer cells depolymerizing the microtubules [31]. Lyngbyabellins E–I are cyclic depsipeptides which disrupt actin in Neuro-2a cells with IC_50s_ 0.7–1.8 μM [37].

STAT3 suppression induces apoptosis and inhibits metastasis in cancer cells. MMP2 and MMP9 are upregulated when the STAT3 pathway is activated, facilitating tumor invasion [97]. Apratoxin A is proposed to inhibit the phosphorylation of the signal transducer and activator of transcription (STAT) 3, causing metastasis in Neuro-2a cells with an IC_50_ of 1 µM [42,98].

Proteases are critical signaling molecules engaged in a variety of key processes such as apoptosis, metastasis and angiogenesis [99,100,101]. Serine proteases are highly expressed in NB [102]. Numerous cyanobacterial peptides have been shown to interfere with serine protease functions. Symplocamide A has been reported to induce cytotoxicity in Neuro-2a cells [40,103]. The inhibition of chymotrypsin (IC_50_ of 0.38 μM) and trypsin (IC_50_ of 80.2 μM) by this compound further supports its antimetastatic effects [40].

### 3.5. Antiangiogenic Effect

VEGF and MMPs play an essential role in angiogenesis [104,105]. Angiogenesis is believed to be a fundamental prerequisite for the development, invasion, and metastasis of malignant NBs. Anti-angiogenic agents that inhibit neovascularization could represent a potential therapeutic strategy for NB [106]. The marine peptide coibamide A inhibits cancer cell migration by lowering the VEGFR2 and MMP-9 expressions [57,58].

### 3.6. Autophagy

In the early stages of cancer, autophagy functions as a barrier to protect cells against damaging stimuli and malignant development [107,108]. The activation of mTOR inhibits autophagy induction and promotes tumor growth and metastasis (Figure 3). Therefore, the regulation of autophagy with mTOR inhibitors provides an anticancer effect [109]. AMPK activates the autophagy-initiating kinase Ulk1 and phosphorylates TSC2. TSC2 activation can inhibit the mTOR complex 1 (mTORC1), thus promoting autophagy [110]. Acyclolaxaphycins B and B3 increase AMPK phosphorylation and induce mTOR inhibition in SH-SY5Y cells with an IC_50_ of 10 µM [61].

### 3.7. Unknown Mechanisms for Anticancer Activity

Several peptides including floridamide [55], guineamides B–C and G [32,33], hermitamides A–B [45], hoiamide A [36], jamaicamides A–C [46], and tiahuramides B–C [34], isolated from *Lyngbya majuscula*, bouillonamide [42], ulongamide A [42], isolated from *Lyngbya bouillonii* (now called *Moorea bouillonii*) [111]; wewakpeptin A–D [56] isolated from *Lyngbya semiplena*; dragonamides C and D [49], microcolin A–B, and desacetylmicrocolin B [50] isolated from *Lyngbya polychroa* all display significant cytotoxicity, though their specific modes of action have yet to be characterized.

## 4. Clinical Trial Status

Clinical trials for several anticancer substances derived from marine cyanobacteria and microalgae are underway [112]. The U.S. Food and Drug Administration (FDA) approved the use of the marine peptide-derived drug, brentuximab vedotin (marketed as Adcetris), for the treatment of cancer in 2011 [113]. Enfortumab vedotin, Glembatumumab vedotin, Tisotumab vedotin, and other derivatives in the form of antibody–drug conjugates are in Phase III clinical research, while ABBV-085, ASG-15ME, and AGS-67E are in Phase I of clinical research for various types of malignancies (www.clinicaltrials.gov). Soblidotin (also known as TZT-1027, auristatin PE), a dolastatin 10 derivative, has demonstrated potential in the treatment of human colon cancer and has moved into phase II clinical trials [114]. Dolastatin 10, ET-743, and bryostatin 1 are currently being evaluated in clinical research [115]. ILX-651 did not show any cardiovascular toxicity, unlike other dolastatins. Phase I and Phase II of clinical research have been successfully completed, and ILX-651 has been proven to be safe and highly tolerated [116]. Dolastatins, promising molecules for solid tumors, have yet to enter Phase III studies [117]. Phase II clinical studies for Tasidotin, Synthadotin (ILX-651) are also being conducted by Genzyme Corporation and are generated from a marine bacterium (Cambridge, MA, USA). Phase III clinical studies for Soblidotin (TZT 1027), a different bacterial peptide of marine origin, are being conducted by Aska Pharmaceuticals (Tokyo, Japan). These two substances are both promising potential anticancer strategies [118].

## 5. Conclusions and Future Prospective

Neuroblastoma is the most common and deadliest childhood disease. Existing therapies are effective, but they have adverse effects, and relapses are common, highlighting the need for innovative cancer treatments [119]. Cyanobacteria afford an excellent source of metabolites for anticancer drug discovery. Their inexpensive cultivation has enhanced their application in therapeutic development. However, limitations to anti-NB peptide research include a lack of ethnomedicinal basis, technological difficulty in collecting deep-sea marine species, as well as isolation and purification concerns. Modern technology has enabled the collection of marine samples and various peptides from aquatic sources. Moreover, as data in this field are limited, the toxicity and adverse effects of cyanobacterial compounds and metabolites in normal cells must also be assessed [71,120,121,122].

The anti-NB actions of marine cyanobacterial peptides include cell growth suppression, apoptosis induction, cell cycle arrest, sodium channel blockage, antimetastatic activity, and autophagy. These peptides appear to be a powerful and exciting resource for developing anti-NB medicines and a platform for uncovering new therapeutic cellular targets. Insufficient progress has been made, and additional research into the anti-NB mechanisms of marine peptides is needed to generate novel candidate molecules [123].

Most research on anti-NB activity has been carried out in vitro, limiting the transfer of the information for clinical efficacy. The absence of in vivo and clinical studies and an insufficient understanding of the mechanisms of action of marine peptides make them a promising tool for future research. A short half-life, limited bioavailability, processing and manufacturing issues, and sensitivity to proteases are all key limitations for the use of peptides in cancer treatment. D-amino acid substitution, peptide cyclization, encapsulation with nanoparticles, pegylation, and XTEN conjugation can be employed to address metabolic instability and short circulating half-life. D-amino acid substitution also reduces immunogenicity [124,125,126]. Peptides coated with exosomes, liposomes, carbon nanotubes, and dendrimers have significantly improved BBB permeability and thus solve the problem of drug delivery to the brain [127,128,129].

The literature discussed above suggests that, given the current state of many cancer-related disorders, cyanobacterial peptide-based nanoformulated delivery systems immediately need to be commercialized. Commercial nanoformulations or stable and effective drug formulations based on cyanobacterial peptides and their optimal use in nanomedicine-based therapies have not yet been investigated in the literature. Without running the risk of experiencing any negative side effects, it could be interesting to experiment with the use of diatoms and other cyanobacterial peptides species in commercial nanoformulations for the treatment of cancer. The anticancer potential for marine cyanobacterial peptides with nanoformulated medicinal characteristics will be unlocked by research in this field. Therefore, cyanobacterial peptides have a strong potential to become anti-NB medicines. Future research on marine peptides could lead to the discovery of new anti-NB therapies.

## Figures and Tables

**Figure 1 cancers-15-02515-f001:**
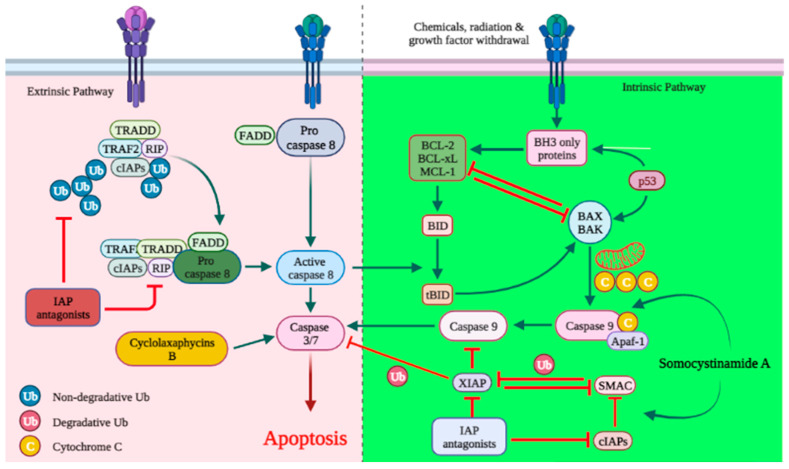
Cyanobacterial metabolites disrupt vital cancer-related pathways. Cyclolaxaphycins B and B3 increase caspase 3 and stimulate apoptosis. Somocystinamide A stimulates caspase-8 activation in Neuro-2a cells. Ub; ubiquitin protein.

**Figure 2 cancers-15-02515-f002:**
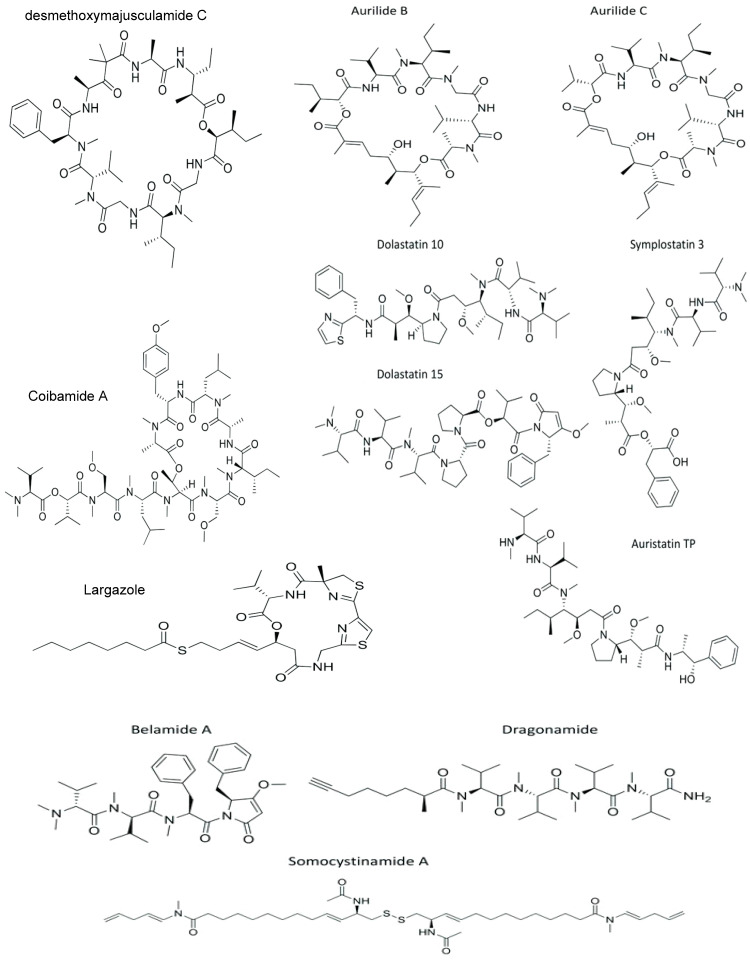
Structures of therapeutically active marine cyanobacterial peptides against neuroblastoma cell lines.

**Figure 3 cancers-15-02515-f003:**
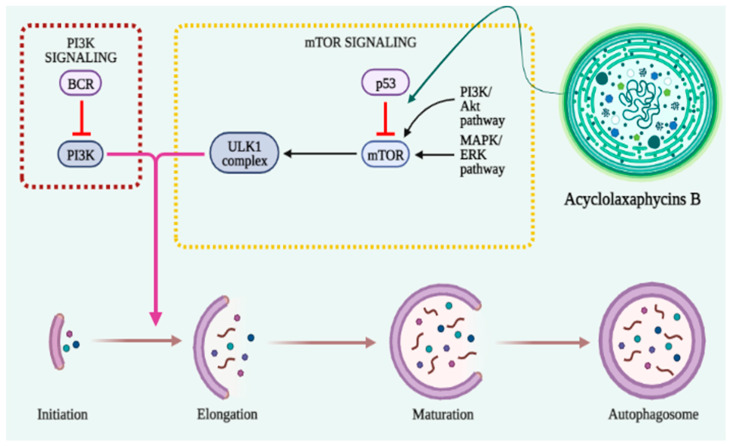
Cyanobacterial peptides involved in the activation of mTOR autophagy.

**Table 1 cancers-15-02515-t001:** Anti-neuroblastoma effects of marine cyanobacterial peptides [53].

Peptides	Cyanobacteria	Active Derivative	Cell Lines	Cytotoxic Concentration IC_50_	Anticancer Mechanisms	References
Aurilide B–C	*Lyngbya majuscula*	cyclic depsipeptide	Neuro-2a	LC_50_ B: 0.01; C: 0.05 μM	PHB1 inhibition; OPA1 proteolysis; microtubule stabilization	[30,54]
Desmethoxymajusculamide C				>1.0 μM	microtubule depolymerization	[31]
Floridamide				EC_50_: 1.89 × 10^−5^ µM mL^−1^	↓ cell viability ^a^	[55]
Guineamide B–C				B: 15; C:16 μM		[32]
Guineamide G				LC_50_: 2.7 μM		[33]
Hoiamide A				2.1 μM		[36]
Lyngbyabellins E–I				LC_50_ E: 1.2, F: 1.8, G: 4.8, H: 1.4, I: 0.7 μM	actin microfilament disruption	[37]
Palmyramide A				17.2 μM	sodium channel blocking activity	[35]
Tiahuramides B–C			SH-SY5Y	B: 14; C: 6 μM	↓ cell viability ^a^	[34]
Apratoxin A	*Lyngbya bouillonii/Moorea bouillonii*		Neuro-2a	1 µM	Stat3 *↓*	[42]
Bouillonamide				6 µM	↓ cell viability ^a^	
Ulongamide A				16 µM		
Wewakpeptin A–D	*Lyngbya semiplena*			A: 0.49; B: 0.20; C: 10.7; D: 1.9 μM		[56]
Coibamide A	*Leptolyngbya* sp.			LC_50_ < 23 nM	caspase-3,7*↑*; cyt c release *↑*; PARP *↑*; VEGFR2 *↓* and MMP-9 *↓*; G1 to S phase arrest	[38,57,58]
Dolastatin 12				>1 μM	G1 and G2/M phase arrest	[39]
Grassypeptolide D and E				D: 599; E: 407 nM		
Ibu-epidemethoxylyngbyastatin 3				>10 μM		
Symplocamide A	*Symploca* sp.			29 nM	antimetastatic (chymotrypsin and trypsin inhibition)	[40]
Largazole			IMR-32	GI_50_: 16 nM	HDACi; MSI ↓; PARP cleavage; G1 and G2/M phase arrest	[41,59,60]
			SH-SY5Y	102 nM		
Cyclolaxaphycins B and B3	*Anabaena torulosa*	cyclic lipopeptide	SH-SY5Y	B: 1.8, B3: 0.8 µM	caspase 3 *↑*	[61]
Acyclolaxaphycins B and B3				10 µM	autophagy (AMPK phosphorylation *↑* and mTOR inhibition)	[61]
Hermitamides A–B	*Lyngbya majuscula*	lipopeptide	Neuro-2a	A: 2.2; B: 5.5 Μm	↓ cell viability ^a^	[45]
Jamaicamides A–C				LC_50_: 15 Μm		[46]
Malyngamide C				LC_50_: 3.1 μg mL^−1^	sodium channel blocking activity	[47]
Malyngamide J				LC_50_: 4 μg mL^−1^		
Malyngamide K				LC_50_: 0.49 μg mL^−1^		
Somocystinamide A				1.4 μg mL^−1^	caspase 8 *↑*	[62]
Somocystinamide A			NB7	810 Nm		[48]
Dragonamide C and D	*Lyngbya* *polychroa*		IMR-32	GI_50_ = C: 49; D: 51 Μm	↓ cell viability ^a^	[49]
Microcolin A–B				A: 0.31; B: 7.7 nM		[50]
Desacetylmicrocolin B		linear peptide		14 nM		
Gallinamide A	*Schizothrix* sp.	linear peptide	Neuro-2a	16.9 µM		[52]

Neuroblastoma Cell lines: Neuro-2a, N-18 = mouse; IMR-32, NB7, SH-SY5Y = human; EC50 = half-maximal effective concentration, GI50 = growth inhibition 50%, and LC50 = lethal concentration 50; ^a^ = mechanism is yet to be investigated, ↑ = increases, ↓ = decreases

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
