# Peer review of "Marine Cyanobacterial Peptides in Neuroblastoma: Search for Better Therapeutic Options"

_cancers, 2023, doi:10.3390/cancers15092515_

Round 1
Reviewer 1 Report
I have to note that the manuscript has not numbered lines, and this made it harder to review and comment on. This goes both to the authors and the editors.
References are missing in several parts, e.g. “Despite the vast array of cancer therapies, current therapies do not deliver the intended results since cancer recurrence and metastasis are prevalent. NB remains a significant problem with a high fatality rate.” or “Numerous prospective studies with marine peptides for pharmaceutical development, including anticancer potential, have been carried out. They have several advantages over pro-teins or antibodies, including small size, simple manufacturing, cell mem-brane crossing capabilities, minimal drug-drug interaction, minimal change in blood-brain barrier (BBB) integrity, selective targeting, chemi-cal and biological diversity, and effects on liver and kidney accumulation” or “These peptides are active against mouse (Neuro-2a, N-18) and Human (IMR-32, NB7 and SH-SY5Y) cell lines” (Introduction). Since this is a Review paper authors must be careful in giving the specific reference to specific information, otherwise the general audience cannot seek the proper information, which is vital especially for review papers.
Nomeclature: Authors should be very careful in cyanobacterial taxa as the nomenclature is constantly updated. For example Lyngbya bouillonii is now called Moorea bouillonii (L.Hoffmann & V.Demoulin) Engene, Rottacker, Kastovsky, Byrum, Choi, Ellisman, Komárek & Gerwick, 2012 (see the Engene et al. 2012 Int J Syst Evol Microbiol. 2012 May; 62(Pt 5): 1171–1178.) and several members of marine Leptolyngbya are now part of the genus Leptothoe (see Konstantinou et al 2019 J. Phycol., Konstantinou et al 2021 Marine Drugs). Same goes with Lynbgya polychroa, Phormidium gracile etc. Please check all taxa carefully and make the necessary corrections, as it is vital for the community which taxa are candidate as drug sources.
International Cooperative Biodiversity Group (ICGB) program: please give References or explain. Is this the program that was terminated already in 2001??
Again, “The majority of the cyclic peptides found in marine cyanobacteria are cyclic depsipeptides, which include 76 different molecules [58]”, this information does not correspond to the paper [58]. This brings up an issue of creditability of the sources, which is vital, especially for a review paper. Authros should check carefully and update their sources, before this manuscript can be published.
“Through the cytotoxicity-directed isolation of a marine cyanobacterium Symploca cf. hydnoides sample from Cetti Bay, Guam, seven novel cyclic hexadepsipeptides, known as veraguamides A-G, were discovered”. Another information without source.
Lyngbya majuscule vs Lyngbya majuscula: again, proper nomenclature is critical
Figure 1 needs improvement. For example Ub is not explained in the legend or the abbreviations. Also, this complex Figure is poorly connected to the text.
“Numerous cyanobacterial peptides have been shown to impede serine protease function. Symplocamide A has been reported to induce cytotoxicity in Neuro-2a”. I cannot see the link of Symplocamide A with proteases and no Reference is given.
Figure 2. No link to the text. Further, the mechanism described in the text (“AMPK activates au-tophagy-initiating kinase Ulk1 and phosphorylate TSC2. TSC2 activation can inhibit mTOR complex 1 (mTORC1), thus promoting autophagy [103]. Acyclolaxaphycins B and B3 increase AMPK phosphorylation and induce mTOR inhibition in SH-SY5Y cells with IC50 of 10μM”) is not illustrated in Figure 3. Why?
Section 3 Clinical Trial Status: Most of the references given there are not up-to-date and the general link to https://www.clinicaltrials.gov/ is not helping the reader. Please improve this part.
Table 1 is not linked to the text. For a comprehensive and accurate list of compounds I strongly recommend that the authors checked the CyanoMetDB database (Jones et al. 2021 Water Research 196 117017)
Author Response
Reviewer 1
Comments and Suggestions for Authors
I have to note that the manuscript has not numbered lines, and this made it harder to review and comment on. This goes both to the authors and the editors.
References are missing in several parts, e.g. “Despite the vast array of cancer therapies, current therapies do not deliver the intended results since cancer recurrence and metastasis are prevalent. NB remains a significant problem with a high fatality rate.” or “Numerous prospective studies with marine peptides for pharmaceutical development, including anticancer potential, have been carried out. They have several advantages over pro-teins or antibodies, including small size, simple manufacturing, cell mem-brane crossing capabilities, minimal drug-drug interaction, minimal change in blood-brain barrier (BBB) integrity, selective targeting, chemi-cal and biological diversity, and effects on liver and kidney accumulation” or “These peptides are active against mouse (Neuro-2a, N-18) and Human (IMR-32, NB7 and SH-SY5Y) cell lines” (Introduction). Since this is a Review paper authors must be careful in giving the specific reference to specific information, otherwise the general audience cannot seek the proper information, which is vital especially for review papers.
Reply: Thank you very much for the valuable comments. The needful suggested sections have been cited and highlighted.
Nomeclature: Authors should be very careful in cyanobacterial taxa as the nomenclature is constantly updated. For example Lyngbya bouillonii is now called Moorea bouillonii (L.Hoffmann & V.Demoulin) Engene, Rottacker, Kastovsky, Byrum, Choi, Ellisman, Komárek & Gerwick, 2012 (see the Engene et al. 2012 Int J Syst Evol Microbiol. 2012 May; 62(Pt 5): 1171–1178.) and several members of marine Leptolyngbya are now part of the genus Leptothoe (see Konstantinou et al 2019 J. Phycol., Konstantinou et al 2021 Marine Drugs). Same goes with Lynbgya polychroa, Phormidium gracile etc. Please check all taxa carefully and make the necessary corrections, as it is vital for the community which taxa are candidate as drug sources.
Reply: The respective section is updated and suggested references are cited.
International Cooperative Biodiversity Group (ICGB) program: please give References or explain. Is this the program that was terminated already in 2001??
Reply: The respective section is cited
Again, “The majority of the cyclic peptides found in marine cyanobacteria are cyclic depsipeptides, which include 76 different molecules [58]”, this information does not correspond to the paper [58]. This brings up an issue of creditability of the sources, which is vital, especially for a review paper. Authros should check carefully and update their sources, before this manuscript can be published.
Reply: The reference is updated and highlighted
“Through the cytotoxicity-directed isolation of a marine cyanobacterium Symploca cf. hydnoides sample from Cetti Bay, Guam, seven novel cyclic hexadepsipeptides, known as veraguamides A-G, were discovered”. Another information without source.
Reply: References have been cited
Lyngbya majuscule vs Lyngbya majuscula: again, proper nomenclature is critical
Reply: corrected
Figure 1 needs improvement. For example Ub is not explained in the legend or the abbreviations. Also, this complex Figure is poorly connected to the text.
Reply: updated and revised
“Numerous cyanobacterial peptides have been shown to impede serine protease function. Symplocamide A has been reported to induce cytotoxicity in Neuro-2a”. I cannot see the link of and no Reference is given.
Reply: this section is updated and revised
Figure 2. No link to the text. Further, the mechanism described in the text (“AMPK activates au-tophagy-initiating kinase Ulk1 and phosphorylate TSC2. TSC2 activation can inhibit mTOR complex 1 (mTORC1), thus promoting autophagy [103]. Acyclolaxaphycins B and B3 increase AMPK phosphorylation and induce mTOR inhibition in SH-SY5Y cells with IC50 of 10μM”) is not illustrated in Figure 3. Why?\
Reply: figure is updated
Section 3 Clinical Trial Status: Most of the references given there are not up-to-date and the general link to https://www.clinicaltrials.gov/ is not helping the reader. Please improve this part.
Reply: This section is updated with fresh references moreover https://www.clinicaltrials.gov/ has updated list.
Table 1 is not linked to the text. For a comprehensive and accurate list of compounds I strongly recommend that the authors checked the Cyano MetDB database (Jones et al. 2021 Water Research 196 117017)
Reply: The respective section is updated and cited with the suggested references.
Regards
Reviewer 2 Report
The search for new ways to treat malignant brain tumors is one of the most serious tasks of contemporary medicine, including pediatrics, since mortality from this type of tumor is very high.
Marine organisms – animals, plants, protozoa - are considering as a promising source of various pharmaceuticals, including antitumor. Therefore, the topic of this manuscript is highly relevant.
The manuscript summarizes the literature data on numerous peptides of cyanobacteria belonging to several structural types, including linear, cyclic, lipopeptides, etc. with a variety of antitumor mechanisms of action. The review contains a list of known anticancer peptides of marine cyanobacteria and attempts to systematize them by mechanisms of action, with a focus on peptides acting on cultured mouse cells of the Neuro-2a and N-18 and human lines (IMR-32, NB7 and SH-SY5Y), which can be tested for the treatment of neuroblastoma.
In general, the manuscript is well-written, but there are some misprints and unclear points.
I have the following comments:
In the chapter "3. Mechanical insights", the authors use the wrong numeration of sections: 2.1. Apoptosis; 2.2. Cell Cycle Arrest; etc. Obviously, the numbering of sections starting from “2” is erroneous, and the sections should be renumbered, for example, "3.1. Apoptosis ", etc. The numbering of the chapters is also confusing – two chapters have the same number “3” – “3. Mechanical insights” and “3. Clinical Trial Status”. It must be corrected.
The molecular mechanisms of the antitumor action of the peptides in some cases are unclear. I recommend adding more information to clarify these points.
For example, it is written: “Coibamide A induces apoptosis in Neuro-2a (LC50 < 23nM) by caspase-3,7 activations, cyt-c release and PARP cleaved [38, 68, 69]”. Are there any assumption what the certain protein (or its specific domain) is the main target of the peptide’s action? What does it mean: “PARP cleaved”? Is it PARP 1 or all PARPs? What enzyme is involved in this cleavage (or ADP-deribosylation?)?
Another example: “Somo-cystinamide A stimulates caspase-8 activation in Neuro-2a and NB7 cells with IC50s of 1.4 μg mL−1 and 810nM [48, 72]”. What is IC50 in this case? The beginning of this phrase indicates the stimulating activity. If IC50 is related to cell viability, it should be written more clearly.
In general, I recommend explaining in more detail the supposed protein-protein interactions that may underlie the observed effects of the cyanobacterial peptides: blocking key signaling pathways, reducing the functional activity of various proteins, etc. If this is not currently known, it is better to mention that it requires further research.
I also suggest adding a few suggestions regarding the effects of peptides on normal brain cells, proving the absence of their toxicity. Otherwise, it is impossible to recommend them as promising antitumor drugs.
Author Response
Reviewer 2
Comments and Suggestions for Authors
The search for new ways to treat malignant brain tumors is one of the most serious tasks of contemporary medicine, including pediatrics, since mortality from this type of tumor is very high.
Marine organisms – animals, plants, protozoa - are considering as a promising source of various pharmaceuticals, including antitumor. Therefore, the topic of this manuscript is highly relevant.
The manuscript summarizes the literature data on numerous peptides of cyanobacteria belonging to several structural types, including linear, cyclic, lipopeptides, etc. with a variety of antitumor mechanisms of action. The review contains a list of known anticancer peptides of marine cyanobacteria and attempts to systematize them by mechanisms of action, with a focus on peptides acting on cultured mouse cells of the Neuro-2a and N-18 and human lines (IMR-32, NB7 and SH-SY5Y), which can be tested for the treatment of neuroblastoma.
In general, the manuscript is well-written, but there are some misprints and unclear points.
I have the following comments:
In the chapter "3. Mechanical insights", the authors use the wrong numeration of sections: 2.1. Apoptosis; 2.2. Cell Cycle Arrest; etc. Obviously, the numbering of sections starting from “2” is erroneous, and the sections should be renumbered, for example, "3.1. Apoptosis ", etc. The numbering of the chapters is also confusing – two chapters have the same number “3” – “3. Mechanical insights” and “3. Clinical Trial Status”. It must be corrected.
Reply: the suggested changes have been done and manuscript is revised
The molecular mechanisms of the antitumor action of the peptides in some cases are unclear. I recommend adding more information to clarify these points.
For example, it is written: “Coibamide A induces apoptosis in Neuro-2a (LC50 < 23nM) by caspase-3,7 activations, cyt-c release and PARP cleaved [38, 68, 69]”. Are there any assumption what the certain protein (or its specific domain) is the main target of the peptide’s action? What does it mean: “PARP cleaved”? Is it PARP 1 or all PARPs? What enzyme is involved in this cleavage (or ADP-deribosylation?)?
Reply: the suggested changes have been done and highlighted.
Another example: “Somo-cystinamide A stimulates caspase-8 activation in Neuro-2a and NB7 cells with IC50s of 1.4 μg mL−1 and 810nM [48, 72]”. What is IC50 in this case? The beginning of this phrase indicates the stimulating activity. If IC50 is related to cell viability, it should be written more clearly.
Reply: the section is updated and highlighted
In general, I recommend explaining in more detail the supposed protein-protein interactions that may underlie the observed effects of the cyanobacterial peptides: blocking key signaling pathways, reducing the functional activity of various proteins, etc. If this is not currently known, it is better to mention that it requires further research.
Reply: up till now it is not known and further research is going on
I also suggest adding a few suggestions regarding the effects of peptides on normal brain cells, proving the absence of their toxicity. Otherwise, it is impossible to recommend them as promising antitumor drugs.
Reply: this section is updated and highlighted.
Regards
Reviewer 3 Report
Marine Cyanobacterial Peptides in Neuroblastoma: Search for Better Therapeutic Option is promising. However, English is substandard.
I couldn’t find the Marine Cyanobacterial Peptides' molecular structure.
The introduction should be supplemented through a hypothesis.
Figures should be illustrated in high resolution.
The Abstract and conclusion section needs to be rewritten. Therefore, I recommended it for major revision.
Author Response
Reviewer 3
Comments and Suggestions for Authors
Marine Cyanobacterial Peptides in Neuroblastoma: Search for Better Therapeutic Option is promising. However, English is substandard.
I couldn’t find the Marine Cyanobacterial Peptides' molecular structure.
Reply: the needful changes have been done
The introduction should be supplemented through a hypothesis.
Reply: the needful suggested changes have been done
Figures should be illustrated in high resolution.
Reply: high resolution figures are added.
The Abstract and conclusion section needs to be rewritten. Therefore, I recommended it for major revision.
Reply: abstract and conclusion section is updated.
Regards
Round 2
Reviewer 3 Report
The author did the great work.
Author Response
Thanks for the acceptance of effort